# Evaluating Object-Centric Models beyond Object Discovery

**Krishnakant Singh** [1]   **Simone Schaub-Meyer** [1 2 3]   **Stefan Roth** [1 2 3]

## Abstract

Object-centric learning (OCL) aims to learn structured scene representations that support compositional generalization and robustness to out-of-distribution (OOD) data. However, OCL models are often not evaluated regarding these goals. Instead, most prior work focuses on evaluating OCL models solely through object discovery and simple reasoning tasks, such as probing the representation via image classification. We identify two limitations in existing benchmarks: *(1)* They provide limited insights on the representation usefulness of OCL models, and *(2)* localization and representation usefulness are assessed using disjoint metrics. To address *(1)*, we use instruction-tuned VLMs as evaluators, enabling scalable benchmarking across diverse VQA datasets to measure how well VLMs leverage OCL representations for complex reasoning tasks. To address *(2)*, we introduce a unified evaluation task and metric that jointly assess localization (*where*) and representation usefulness (*what*), thereby eliminating inconsistencies introduced by disjoint evaluation. Finally, we introduce a simple multi-feature reconstruction baseline that outperforms existing OCL methods on several benchmarks. *Project page and code:* https://visinf.github.io/byod

## 1. Introduction

Object-centric learning (OCL) aims at decomposing a scene into a set of latent object representations. Thereby, OCL methods aim to enable vision systems to reason about scenes by representing them as sets of constituent objects, akin to how humans reason about the world (Baillargeon et al., 1985; Spelke, 1990; Téglás et al., 2011). Reasoning at the

[1]Department of Computer Science, TU Darmstadt [2]hessian.AI [3]Zuse School ELIZA. Correspondence to: Krishnakant Singh <krishnakant.singh@visinf.tu-darmstadt.de>.

*Proceedings of the 43rd International Conference on Machine Learning*, Seoul, South Korea. PMLR 306, 2026. Copyright 2026 by the author(s).

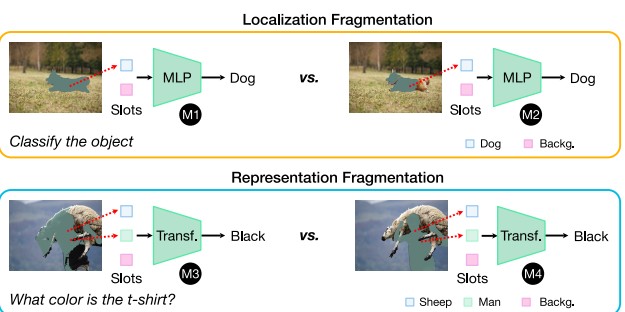

*Figure 1.* **Limitations of the disjoint evaluation of OCL models.** Disjoint metrics ignore localization and representation fragmentation. *(top)* Models M1 and M2 obtain the same classification score despite M1 localizing the object more accurately. We term this issue in evaluation a localization fragmentation issue. *(bottom)* Transformer-based probing for VQA tasks does not attribute answers to specific slots; hence, model M4 (correct answer from correct slot) is scored the same as M3, which answers correctly albeit using the wrong slot. We term this evaluation issue a representation fragmentation issue.

level of objects is thought to enable compositional or systematic generalization (Greff et al., 2020; Wiedemer et al., 2024; Kapl et al., 2025), improve robustness to out-of-distribution (OOD) samples (Dittadi et al., 2022; Arefin et al., 2024), and support causal reasoning (Schölkopf et al., 2021; Mansouri et al., 2024). Among various OCL approaches (Greff et al., 2019; Engelcke et al., 2020; Lin et al., 2020), slot attention-based methods (Locatello et al., 2020) have gained popularity for their strong performance on real-world data (Everingham et al., 2010; Lin et al., 2014).

Existing evaluation benchmarks for OCL suffer from two key limitations: *(1) Limited evaluation on complex reasoning tasks.* Most OCL models are typically evaluated on an unsupervised object discovery task. This serves as a poor proxy (Rubinstein et al., 2025) for evaluating the broader goals behind OCL models, such as OOD generalization, counterfactual reasoning, and compositional generalization to novel scenes. Using the common linear probing for each such capability is cumbersome and does not scale, as it requires repeated retraining to evaluate each capability. Moreover, some qualities (*e.g.*, counterfactual reasoning) are not naturally expressed as small closed-set classification problems, making linear probing ill-suited and potentially ambiguous. Other proposed solutions for evaluating OCL performance on visual question-answering tasks (Mamaghan

et al., 2025) fall short, as they also require retraining to assess different reasoning abilities of OCL models. *(2) Disjoint evaluation metrics.* Localization and representation usefulness are typically evaluated using separate metrics. This disjoint evaluation can lead to inconsistencies such as *localization fragmentation*, where a model may capture semantics well but fail to localize the object correctly, and *representation fragmentation*, where a single object is encoded by multiple slots (see Fig. 1).

To address *Limitation 1*, we propose a scalable evaluation framework for OCL that leverages visual instruction tuning to convert a large language model (LLM) into a vision-language model (VLM) and uses an object-centric model as the vision encoder. This enables a *zero-shot evaluation* using diverse visual question-answering (VQA) benchmarks, designed to test broad visual reasoning capabilities without task-specific training for each new benchmark. Importantly, this evaluation measures how well a VLM leverages object-centric representations across visual reasoning tasks, serving as a practical proxy for the representations' utility. Using VQA-based evaluation alone is not yet sufficient, as it does not account for the issue of disjoint assessment of localization and representation usefulness. To address *Limitation 2*, we introduce a unified evaluation task using our enhanced version of the GQA dataset (Hudson & Manning, 2019) and a novel attribution-aware grounded accuracy (AwGA) metric that jointly assesses the object localization and the usefulness of OCL models' representations.

To summarize, our contributions are: *(i)* We benchmark multiple OCL methods by evaluating their usefulness in a VLM across diverse VQA benchmarks designed to test broad visual reasoning abilities; we do so in a zero-shot fashion without requiring retraining. *(ii)* We introduce attribution-aware grounded accuracy (AwGA), a metric that jointly evaluates the "what" and "where" properties of OCL models, addressing localization and representation fragmentation. *(iii)* We show that our VLM-based evaluation and AwGA yield consistent model rankings across different LLM backbones and connectors. *(iv)* We show that multi-feature reconstruction consistently improves the usefulness of object-centric representations under our VLM-based evaluation.

## 2. Related work

**Object-centric learning (OCL)** shares certain goals with other object representation learning approaches, such as CLIP (Radford et al., 2021), DINO (Caron et al., 2020; Oquab et al., 2024), and VQ-VAE (Van Den Oord et al., 2017). However, unlike them, OCL methods aim to learn *object-level* latent representations that enable robust and compositional scene understanding (Dittadi et al., 2022; Wiedemer et al., 2024). Early OCL models relied on VAE-based architectures (Burgess et al., 2019; Greff et al., 2019)

*Table 1.* **Issues with existing evaluation protocols.** Linear and transformer-based probes require retraining for each property, leading to high amortized cost (cost per evaluation), and linear probes are not applicable to tasks requiring open-ended outputs (Open eval.). VLM probes enable diverse evaluations in a zero-shot manner but still suffer from issues of disjoint evaluation (Rep. and Loc. fragmentation). Combining AwGA with VLMs provides a unified evaluation that penalizes OCL methods for representation and location fragmentations.

| Evaluation Protocol | Amort. cost | Open eval. | Rep. frag. | Loc. frag. |
|---|---|---|---|---|
| Linear probes | High | ✗ | ✗ | ✗ |
| Transformer probes | High | ✓ | ✗ | ✗ |
| VLM probes *(ours)* | Low | ✓ | ✗ | ✗ |
| VLM probes + AwGA *(ours)* | Low | ✓ | ✓ | ✓ |

and suffered from scalability issues, which slot attention (Locatello et al., 2020) addressed via iterative attention-based clustering. Other works, such as (Traub et al., 2023), propose methods to explicitly disentangle properties and locations for learning superior slot representations; however, they remain limited to synthetic scenes. Seitzer et al. (2023) were the first to extend slot attention to real images by reconstructing DINO features (Caron et al., 2020). Modern OCL models can be grouped by reconstruction target: image-based approaches (Jiang et al., 2023; Wu et al., 2023; Akan & Yemez, 2025; Singh et al., 2025) reconstruct pixels using strong decoders such as StableDiffusion (Rombach et al., 2022), while feature-based models (Seitzer et al., 2023; Kakogeorgiou et al., 2024; Kim et al., 2024) reconstruct self-supervised encoder features. For a more detailed review of these methods, please see Yuan et al. (2023). In our work, we focus only on slot-attention methods that perform well on real-world images.

**Evaluation of OCL models.** Unsupervised object discovery (UOD) is the most common evaluation for OCL. However, as Rubinstein et al. (2025) argue, UOD is a poor proxy for evaluating OCL models since it does not assess key goals such as compositional generation, counterfactual reasoning, and OOD generalization. Beyond UOD, prior work often evaluates OCL representations via linear probing on downstream property prediction tasks (Locatello et al., 2020; Jiang et al., 2023; Singh et al., 2025), but this setup is ill-suited for complex, varied reasoning tasks: *(i)* it incurs high amortized cost, as each new benchmarking task requires retraining the probes, and *(ii)* many desirable qualities (*e.g.*, counterfactual reasoning) are difficult to express as closed-set classification tasks. To address this, Mamaghan et al. (2025) proposed using a VQA-based evaluation with transformer probes. However, their approach still requires expensive retraining to benchmark each new capability (*e.g.*, compositional generalization; Kapl et al., 2025), resulting in a high amortized cost. Moreover, repeated training for different benchmarks requires a cumbersome hyperparameter search (see Table 1). Thus, we propose an evaluation

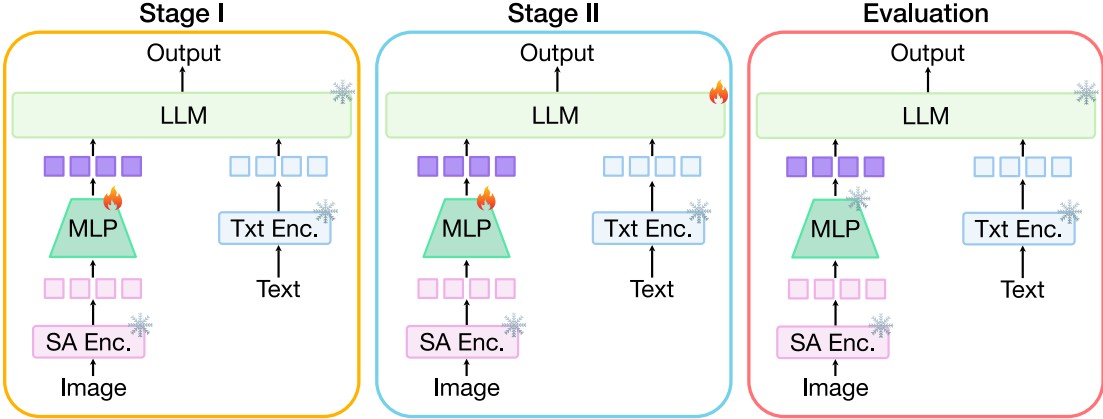

*Figure 2.* **Training and evaluation setup.** Our training is akin to LLaVA (Liu et al., 2023). In *Stage I*, only the MLP connector is trained on the pre-training dataset. This aligns the slot embeddings with the language model's embedding space. In *Stage II*, the MLP network and the language model are trained on the instruction-tuning dataset from LLaVA. This enables the language model to follow instructions and perform tasks based on slots as visual tokens. *Evaluation* is performed in a zero-shot fashion on various VQA benchmarks, where the text is encoded via a text encoder, and images are encoded using the slot-attention model. The evaluation tests how well the connector and LLM networks utilize the slot embeddings for answering the provided questions.

protocol that, once trained, can be reused and extended to cover multiple benchmarks, thereby reducing its amortized cost as more benchmarks are added. Such a protocol can also support evaluation on difficult tasks where collecting sufficient training data for task-specific probes is hard.

Inspired by Tong et al. (2024), we use instruction-tuned vision-language models (VLMs) as probes, enabling *scalable zero-shot evaluation*. Unlike Mamaghan et al. (2025), our work does not require training for each new property evaluation. In particular, our protocol measures how effectively object-centric representations can be aligned with and exploited by a language model across diverse tasks, providing a practical measure of their downstream utility in multimodal systems. However, VQA-based protocols typically evaluate representation and localization separately, overlooking fragmentation effects. We address this by proposing a new evaluation protocol that uses a novel attribution-aware grounded accuracy (AwGA) metric to jointly evaluate localization and representation (see Fig. 1 and Table 1).

## 3. Benchmarking beyond object discovery

### 3.1. Preliminaries

**Slot attention** (**SA**; Locatello et al., 2020) is an iterative refinement framework that decomposes an image into a set of object-centric slots. Given an encoder feature map $\mathbf{H}$, the slot-attention module groups it into $k$ slot vectors $\mathbf{S} = \{\mathbf{s}_1, \ldots, \mathbf{s}_k\}$. At each iteration $t$, slots $\mathbf{S}^t$ are updated via dot-product attention (Vaswani et al., 2017) between the previous slots $\mathbf{S}^{t-1}$ and features $\mathbf{H}$, where the softmax is taken over slots (instead of keys), inducing competition and binding of slots to objects. Typically, SA-based methods use $k = 7$ slots for real-world scenes (Lin et al., 2014), as this

setting performs well on common downstream tasks.

### 3.2. Using VLMs as evaluators

One of the core contributions of our work is to benchmark OCL models by measuring their utility in a VLM across diverse VQA benchmarks designed to test broad visual reasoning capabilities. Extending existing protocols, such as linear probing and transformer-based probing (Mamaghan et al., 2025), requires training probes from scratch for each benchmark (see Table 1). To alleviate this issue, we take inspiration from Tong et al. (2024) and use instruction-tuned VLMs as evaluators, replacing the standard vision encoder (*e.g.*, CLIP (Radford et al., 2021) or DINOv2 (Oquab et al., 2024)) with the OCL encoders.

Our VLM-based evaluation scheme can be written as a function composition $f(g(\mathbf{S}))$, where $f$ denotes an LLM and $g$ denotes a connector network (*e.g.*, 2-layer MLP) that connects the LLM to the slot representation $\mathbf{S}$. Previous evaluation protocols such as linear probing ($f = \mathbf{I}$, $g = 1$-layer MLP) and transformer-based probes ($f = \mathbf{I}$, $g = n$-layer transformer; Mamaghan et al., 2025), where $\mathbf{I}$ denotes the identity function, are special cases of our generalized evaluation protocol. Since both $g$ and $f$ are trained, our benchmark measures the *usefulness* of OCL representations on diverse VQA benchmarks in a zero-shot fashion without requiring retraining for each task. The performance of the OCL model is influenced not only by the visual information in the object-centric representation but also by the ease with which a VLM can align and exploit it to answer textual questions. Note that training linear probes or transformer-based probes similarly evaluate not only what information is present in the representation, but also how easily it can be extracted by the chosen probe class. Our

VLM-based protocol follows the same principle, but uses a stronger probe (an instruction-tuned VLM), enabling scalable evaluation across diverse VQA tasks. We follow the architecture and training protocol of LLaVA (Liu et al., 2023) for learning a vision-language model, using object-centric models as vision encoders.

The training process (see Fig. 2) has two stages: *(i)* In *Stage I*, the slot embeddings are projected by a connector network to align with the space of text embeddings. The text (questions) is tokenized and embedded using the LLM's embedder module. Only the connector network is trained for one epoch on the LLaVA 558K pre-training dataset in this stage. *(ii)* In *Stage II*, both the LLM and connector networks are trained with the LLaVA 665K instruction tuning dataset, which comprises multimodal samples created via GPT-4's responses (Achiam et al., 2023) to images. For more details, see (Liu et al., 2023). The instruction-tuning phase helps the model follow instructions more reliably and improves the VLM's ability to follow instructions and leverage the slot encodings to accurately respond to user prompts.

### 3.3. Joint evaluation protocol – Unifying 'what' and 'where'

With VLM probes, we can evaluate the usefulness of many object-centric models on a diverse set of VQA-based benchmarks *in a zero-shot manner at evaluation time*, without training task-specific probes for each benchmark. This allows us to assess how well slot representations support a broad range of capabilities for which OCL models were originally proposed. However, relying solely on VQA-based evaluation introduces both localization and representation fragmentation (see Fig. 1). Thus, a need exists for a novel evaluation protocol that jointly evaluates and penalizes localization and representation fragmentations. This requires access to a dataset with grounding masks, composed of the masks of all objects required to answer a question.

A way to account for localization fragmentation when evaluating different models with VQA tasks is to use the grounded accuracy (G-Acc; Hudson & Manning, 2019), which is defined as

$$\text{G-Acc} = \frac{1}{N} \sum_{i=1}^{N} \mathbb{1}(\hat{y} = y) \; \text{mIoU}(\mathcal{A}_{\text{pred}}, \mathcal{G}_{\text{GT}}). \quad (1)$$

Here, $\mathcal{A}_{\text{pred}}$ and $\mathcal{G}_{\text{GT}}$ denote the mask predicted from the slots and the ground-truth (GT) grounding masks. $y$ and $\hat{y}$ denote the GT and predicted answer; $\mathbb{1}$ is the indicator function. G-Acc correctly penalizes localization fragmentation; however, it does not consider which slot was used to answer the question. Specifically, G-Acc does not penalize representation fragmentation, *i.e.*, when a model distributes an object's representation across multiple slots (see Fig. 3).

To resolve this and enable the joint evaluation of localiza-

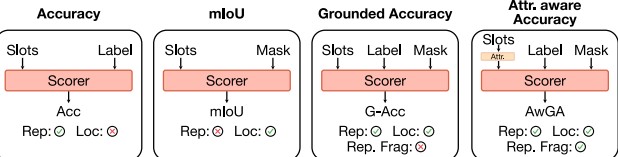

*Figure 3.* **Metrics for evaluating OCL models.** Accuracy and mIoU evaluate representation usefulness and localization separately, while grounded accuracy allows for a joint evaluation, addressing localization fragmentation but overlooks representation fragmentation. Our proposed AwGA jointly evaluates both and penalizes both fragmentation types.

tion and representation usefulness, we propose *AwGA*, an attribution-aware grounded accuracy metric that penalizes a model for committing both localization and representation fragmentation (Fig. 3). Our AwGA metric first computes an attribution score for each slot with respect to the predicted answer (Simonyan et al., 2014). We then select the $K$ slots with the highest attributions and compute the mean intersection over union (mIoU) using the *union* of their predicted masks. For each question, $K$ is set to the number of objects in the grounding mask and is *not* a hyperparameter. This way, the overlap is computed only for the slots that are most responsible for answering the question. AwGA is formally written as

$$\text{AwGA} = \frac{1}{N} \sum_{i=1}^{N} \mathbb{1}(\hat{y} = y) \; \text{mIoU}(\text{TopK}(\mathcal{A}_{\text{pred}}), \mathcal{G}_{\text{GT}}). \quad (2)$$

For computing each attribution, we simply use the gradient of each slot with respect to the loss function (Simonyan et al., 2014; Springenberg et al., 2015). In particular, we compute the sensitivity ($\frac{\partial y}{\partial \mathbf{s}_i}$) of the output $y = f(g(\mathbf{S}))$ with respect to each slot $\mathbf{s}_i$.

## 4. Analysis of OCL methods

**Utility *vs*. grounded faithfulness.** We first benchmark the *utility* of OCL models in a VLM across diverse VQA benchmarks designed for evaluating broad visual reasoning capabilities (Tables 2 and 3). While these evaluations quantify the usefulness of OCL models across diverse complex reasoning tasks, they do not address issues arising from disjoint evaluation (fragmentation failure modes). Additionally, Sec. 4.3 shows that using object discovery as a measure is a poor proxy for an OCL method's downstream capabilities, underscoring the need for a unified metric. We therefore propose a joint evaluation protocol based on our enhanced GQA dataset and using our AwGA metric, simplifying the evaluation of *what* and *where* with a single score (Sec. 4.4).

**Baselines.** The original goal of object-centric learning (OCL) has been to obtain object-centric representations in an *unsupervised* manner; we thus focus our evaluation

on unsupervised OCL methods. We take state-of-the-art baselines for real-world datasets (*e.g.*, COCO; Lin et al., 2014), including SPOT (Kakogeorgiou et al., 2024), Slot-Diffusion (Wu et al., 2023), StableLSD (Jiang et al., 2023), FT-DINOSAUR (Didolkar et al., 2025), and DINOSAUR (Seitzer et al., 2023). Whenever available, we use the authors' released checkpoints; StableLSD and DINOSAUR are retrained from scratch using their official scripts. We also include a DINOSAURv2 baseline, which replaces the original DINO (Caron et al., 2021) backbone with DINOv2 (Oquab et al., 2024) and a vector-quantized version of DINO (VQDINO$_{\text{MLP}}$) introduced by Zhao et al. (2025).

**Improved baseline.** Existing OCL models typically use either feature reconstruction (Seitzer et al., 2023; Kakogeorgiou et al., 2024) or image reconstruction (Jiang et al., 2023; Wu et al., 2023) as their training target. To complement this, we propose a simple improved baseline, mFRESA, which combines multiple reconstruction targets: image pixels, DINOv2 features, and HOG features (Dalal & Triggs, 2005). Concretely, mFRESA builds on StableLSD (Jiang et al., 2023) and adds two lightweight decoders: *(1)* a feature decoder that reconstructs DINOv2 features, akin to Seitzer et al. (2023), and *(2)* a three-layer MLP that reconstructs patch-level HOG features (Dalal & Triggs, 2005). Given the slots, the *HOG decoder* reconstructs the HOG feature map of the input image (Dalal & Triggs, 2005), encouraging slots to better capture object boundaries via edge information. The *DINOv2 feature decoder*, inspired by DINOSAUR (Seitzer et al., 2023), reconstructs DINOv2 features from the slots, complementing image-level supervision. The overall training objective is given as

$$\mathcal{L} = L_2(I_{\text{inp}}, I_{\text{recon}}) + L_2(F_{\text{inp}}, F_{\text{recon}}) + L_2(H_{\text{inp}}, H_{\text{recon}}),$$
(3)

where $I$ denotes images, $F$ DINOv2 features, and $H$ HOG features of the input (inp) and reconstruction (recon), respectively. Fig. 4 shows the detailed architecture for mFRESA.

The key contribution of mFRESA is the *joint reconstruction of image, feature, and edge signals*, enabling slots to learn stronger object-centric representations.

**Training details for VLM-based evaluation of OCL methods.** We use Phi2-3B (Javaheripi et al., 2023) and Qwen2-7B (Yang et al., 2024) as language models in our VLM evaluation setup. We use a 2-layer MLP with GeLU activations (Hendrycks & Gimpel, 2016) as the connector network. Pre-training is performed with a batch size of 256 and a learning rate of $1 \times 10^{-3}$, followed by fine-tuning with a batch size of 128 and a learning rate of $2 \times 10^{-5}$, using AdamW (Loshchilov & Hutter, 2018) throughout. The maximum sequence length is set to 2048 tokens. During evaluation, we follow LLaVA and use greedy decoding (temperature 0, beams 1). All training and architectural settings are held fixed across LLMs and vision encoders,

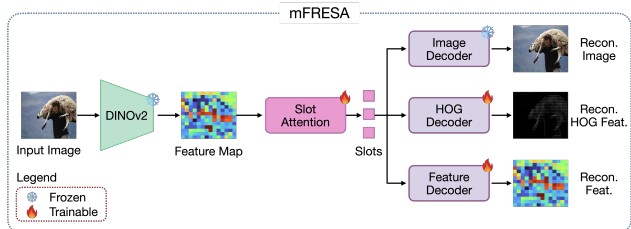

*Figure 4.* **Multi-feature reconstruction for slot attention (mFRESA)** uses a DINOv2 (Oquab et al., 2024) model as a feature encoder network. The slot-attention module groups the obtained features into slots. Multiple decoders reconstruct the image, HOG features, and DINOv2 features from the slots. The slot-attention module and the HOG and feature decoders are trainable, while DINOv2 and the image decoder (a diffusion decoder) are kept frozen, as in StableLSD. The model is trained with Equation (3). Not visualized: The HOG features are computed as described in (Dalal & Triggs, 2005).

ensuring that the only variability comes from the choice of vision encoder (*i.e.*, slot attention) module. VLM training is performed on 8×A100 GPUs (80 GB each). Training time varies across OCL encoders, with the longest runs occurring with mFRESA and StableLSD. For these models, pre-training requires approximately 4 hours, and fine-tuning takes roughly 20–24 hours.

### 4.1. Standard perception evaluation

Using instruction-tuned VLMs as evaluators enables scalable benchmarking of vision encoders across a wide range of tasks represented in a VQA setting. Since our goal is to assess the representational utility of object-centric vision tokens, we focus on image-centric perception benchmarks, including GQA (Hudson & Manning, 2019), VQAv2 (Goyal et al., 2017), MME (Fu et al., 2025), and MM-Vet (Yu et al., 2024). To evaluate whether object-centric encodings can mitigate object hallucination, we also report results on POPE (Li et al., 2023b), which probes object presence via Boolean questions. For MME, we report only perception tasks, as these are most relevant to our setting. To quantify the extent to which performance depends on visual tokens (rather than language priors), we also include a *Blind-VLM* baseline, where vision tokens are replaced with random noise.

As shown in Table 2, despite using far fewer visual tokens (7 *vs.* 196 for DINOv2), OCL models perform competitively with DINOv2, a strong self-supervised vision encoder. Interestingly, FT-DINOSAUR — the leading OCL model for object discovery — underperforms the older DINOSAURv2 on nearly all benchmarks, suggesting that object discovery scores are a poor proxy for the utility of OCL representations (*cf*. Rubinstein et al., 2025) also in our VLM-based visual reasoning benchmarks. Combining reconstruction targets (mFRESA) improves performance across most benchmarks (except MM-Vet), suggesting that multi-target reconstruc-

*Table 2.* **VQA comparison of OCL and foundational models.** A Blind-VLM replaces vision tokens with random noise and serves as a lower bound, isolating the contribution of the vision encoder. DINOv2 provides an upper-bound reference for self-supervised representations, compared against slot-attention models using feature (2[nd] group) or image (3[rd] group) reconstruction. We highlight the **best** and second-best SA models and report VQA accuracy (%, ↑); for MME, only perception tasks are included (↑, max. 2000). Despite using only 7 vision tokens, OCL models remain competitive with DINOv2, which uses 196 tokens. mFRESA, a hybrid feature and image-reconstruction method, outperforms other OCL methods on most benchmarks.

| LLM | Phi2 | | | | | Qwen2-7B | | | | |
|---|---|---|---|---|---|---|---|---|---|---|
| Dataset | GQA | POPE | MME | MMVET | VQAv2 | GQA | POPE | MME | MMVET | VQAv2 |
| Blind-VLM | 38.20 | 64.92 | 667.38 | 13.2 | 44.71 | 40.32 | 65.12 | 686.29 | 15.0 | 45.76 |
| DINOv2 | 57.77 | 82.01 | 1279.21 | 22.6 | 71.15 | 61.83 | 83.31 | 1388.19 | 23.6 | 74.86 |
| DINOSAUR | 49.71 | 78.72 | 1047.34 | 17.5 | 58.41 | 53.63 | 79.54 | 1178.87 | 15.9 | 61.98 |
| DINOSAURv2 | 53.23 | 81.76 | 1122.73 | **18.9** | 63.84 | 56.32 | 82.49 | 1224.84 | 17.7 | 66.23 |
| VQDINO$_{MLP}$ | 52.69 | 81.48 | 1167.08 | 18.8 | 63.22 | 56.2 | 82.36 | 1229.43 | 18.8 | 67.28 |
| FT-DINOSAUR | 52.22 | 81.45 | 1004.94 | 15.1 | 60.97 | 56.15 | 81.85 | 1242.25 | 17.2 | 66.09 |
| SPOT | 51.06 | 79.74 | 1069.80 | 17.4 | 60.94 | 54.94 | 80.24 | 1169.11 | 17.8 | 65.37 |
| Slot Diffusion | 50.00 | 79.77 | 1090.10 | 18.5 | 59.65 | 53.93 | 79.91 | 1171.83 | 18.9 | 63.47 |
| StableLSD | 51.45 | 81.51 | 1129.08 | 17.8 | 62.06 | 55.96 | 81.54 | 1239.48 | 18.6 | 66.67 |
| mFRESA *(ours)* | **53.90** | **82.12** | **1187.05** | 18.5 | **65.58** | **58.28** | **82.74** | **1283.48** | **19.3** | **69.93** |

tion can be a useful design choice for OCL.

> *Takeaway 1.* Under our VLM-based evaluation, OCL models can be competitive with DINOv2 despite using far fewer tokens. Overall, feature-reconstruction models outperform image-reconstruction models, and a hybrid baseline (mFRESA) yields further gains.

### 4.2. Robust perception evaluation

Despite competitive results, OCL models still lag in absolute performance on general perception benchmarks. We next ask whether they offer advantages on tasks where object-centric learning is conjectured to help, such as OOD generalization, compositionality, and counterfactual reasoning (Greff et al., 2020; Wiedemer et al., 2024; Kapl et al., 2025). Using VLM probes enables evaluating OCL encoders *without retraining* across diverse VQA benchmarks designed to test these abilities. Results are shown in Table 3.

*Positives.* On OOD-CV (Tu et al., 2024), which contains images with unusual textures and backgrounds, most OCL models are competitive with DINOv2 despite using far fewer visual tokens, suggesting that OCL models are robust to distribution shifts. We also find that on counterfactual question answering, particularly direct numeric queries (*e.g.*, "How many $X$ would there be if two $X$ were added/removed?"), several OCL models are competitive with or outperform DINOv2. Interestingly, for Boolean counterfactual questions, the non-visual baseline (*Blind-VLM*) performs best. This suggests that many Boolean CVQA questions are structurally simple (*e.g.*, "Would $X$ still be true if $Y$ changed?") and can often be answered using linguistic priors, while adding visual tokens may introduce signals that are not helpful in such cases. It should be noted that success on counterfactual QA does not necessarily imply causal understanding. Also, it can be seen that using a multi-feature

reconstruction target (mostly) improves the performance on these robust perception-reasoning tasks.

*Negatives.* For compositional reasoning, we evaluate Sugar-Crepe (Hsieh et al., 2023), where the model must choose the correct caption between a true caption and a hard-negative caption generated by an LLM (attribute swaps, object additions, or replacements; Achiam et al., 2023). OCL models remain behind DINOv2. Interestingly, for compositional reasoning, our results differ from Kapl et al. (2025) due to evaluation differences: they train OCL models on CLEVR-Tex (Karazija et al., 2021) and compare against frozen DINOv2, whose natural-image pretraining is disadvantaged in this synthetic domain. In contrast, we compare frozen models and evaluate compositional and OOD behavior separately, rather than through a single cOOD split. We also evaluate robustness to natural adversarial examples using NaturalBench (Li et al., 2024), which contains pairs of questions and images designed so that a blind model fails (*i.e.*, the answer changes with the image). Solving Natural-Bench requires object recognition, attribute binding, and relation understanding. We again observe a substantial gap between OCL models and DINOv2, indicating that current OCL representations are less useful for this task in our VLM-based evaluation setting. Additionally, we find that feature-reconstruction-based OCL models outperform image-reconstruction-based models under our VLM probing protocol.

> *Takeaway 2.* Under our VLM-probing evaluation, OCL models are competitive on OOD generalization and numeric counterfactual reasoning, but lag behind DINOv2 on compositional and natural adversarial benchmarks. Feature-reconstruction OCL generally outperforms image-reconstruction on robustness tasks.

*Table 3.* **Robustness of OCL methods.** Evaluation on tasks beyond object discovery, such as OOD generalization, compositional understanding, counterfactual reasoning, *etc.* (accuracy in %, ↑). The Blind-VLM serves as a lower bound while DINOv2 serves as a reference upper bound for performance on these tasks. We highlight the **best** and second best model among SA methods. The datasets evaluate the following properties: CVQA (Zhang et al., 2024) – counterfactual reasoning, OOD-CV (Tu et al., 2024) – OOD generalization, NaturalBench (Li et al., 2024) – robustness to natural adversarial examples, SugarCrepe (Hsieh et al., 2023) – vision-language compositionality.

| LLM | Phi2 | | | | | Qwen2-7B | | | | |
|---|---|---|---|---|---|---|---|---|---|---|
| **Dataset** | CVQA | | OODCV | N. Bench | SugarC. | CVQA | | OODCV | N. Bench | SugarC. |
| | Direct | Boolean | | | | Direct | Boolean | | | |
| Blind-VLM | 28.00 | 71.41 | 50.98 | 0.42 | 49.42 | 32.69 | 65.57 | 51.01 | 0.52 | 53.29 |
| DINOv2 | 36.96 | 63.72 | 58.00 | 8.42 | 82.05 | 45.74 | 53.54 | 58.36 | 9.89 | 88.06 |
| DINOSAUR | 35.74 | 69.29 | 51.97 | 1.89 | 67.85 | 41.13 | 63.72 | 52.52 | 3.95 | 72.45 |
| DINOSAURv2 | 34.52 | 65.75 | 53.90 | 3.37 | 75.98 | 42.09 | **64.07** | 56.66 | 6.16 | 78.18 |
| VQDINO$_{MLP}$ | 35.13 | 66.37 | 52.03 | 3.89 | 73.05 | **42.34** | 55.66 | 53.18 | 6.07 | 80.16 |
| FT-DINOSAUR | **39.13** | 68.85 | 55.18 | 2.89 | 70.94 | 42.00 | 57.17 | 53.28 | 5.42 | 81.24 |
| SPOT | 36.35 | 69.47 | 53.34 | 2.42 | 71.65 | 41.83 | 57.61 | 54.07 | 3.68 | 74.08 |
| Slot Diffusion | 33.83 | 68.23 | 51.34 | 2.21 | 70.39 | 39.39 | 59.56 | 52.56 | 3.74 | 74.53 |
| StableLSD | 38.26 | **70.44** | 52.89 | 3.00 | 72.92 | 41.04 | 62.39 | 55.08 | 5.21 | 78.98 |
| mFRESA *(ours)* | 38.09 | 66.64 | **55.57** | **4.21** | **77.27** | 41.39 | 60.44 | **57.31** | **6.84** | **83.17** |

## 4.3. Are object discovery metrics predictive of downstream utility?

Unsupervised object discovery (UOD) metrics such as mean best overlap (mBO; Pont-Tuset et al., 2016) and mean intersection over union (mIoU) are widely used to evaluate slot-attention methods. Yet, it remains unclear whether higher UOD scores imply greater utility of object representations for diverse reasoning tasks (*cf*. Rubinstein et al., 2025). In Table 4, we compare several OCL methods across UOD metrics, general VQA performance, adversarial robustness, and compositional reasoning. We find that UOD metrics (mIoU and mBO) correlate poorly with the usefulness of slot representations under our VLM-probing evaluation. For instance, FT-DINOSAUR, a leading OCL model for object discovery, performs worse than DINOSAURv2 on general VQA tasks and robustness benchmarks (compositional reasoning and natural adversarial robustness). One plausible explanation is that FT-DINOSAUR finetunes the DINOv2 encoder, whereas other models keep it frozen; finetuning on comparatively small datasets such as COCO can reduce generalization (Mukhoti et al., 2024), which may negatively affect the learned object representations required for reasoning tasks.

> *Takeaway 3.* In our VLM-based evaluation setting, object discovery metrics are weak proxies for the usefulness of object-centric representations, motivating metrics that jointly evaluate localization *and* usefulness of OCL representations.

## 4.4. A joint evaluation protocol

To assess OCL models using our proposed AwGA metric, we use the GQA validation set (Hudson & Manning, 2019), a large-scale VQA dataset with grounding boxes for each question. To better align with our evaluation, we enhance GQA by converting bounding-box annotations into masks using SAM2 (Ravi et al., 2025), treating boxes as prompts. To ensure that grounded objects are salient, we filter out images with more than seven boxes or those covering less than 10% of the image area. We use the same mask-generation pipeline for all models to ensure a fair comparison. We call this dataset the enhanced GQA dataset (eGQA). Some examples are shown in Fig. 5 and more details are provided in Appendix E.

We report accuracy, mIoU, G-Acc, and our proposed AwGA metric in Table 5. mIoU measures the overlap between predicted and ground-truth masks, but by itself, cannot determine whether the correct answer is grounded in the appropriate object slots. G-Acc penalizes poor localization but overlooks fragmented or incorrect slot usage (representation fragmentation). By contrast, AwGA jointly evaluates both localization and representation usefulness under a single VLM probing protocol, providing a grounded measure that penalizes both localization and representation fragmentation (see Fig. 3). Interestingly, models with top object discovery or accuracy scores are not always SOTA under G-Acc or AwGA, underscoring the pitfalls of disjoint evaluation.

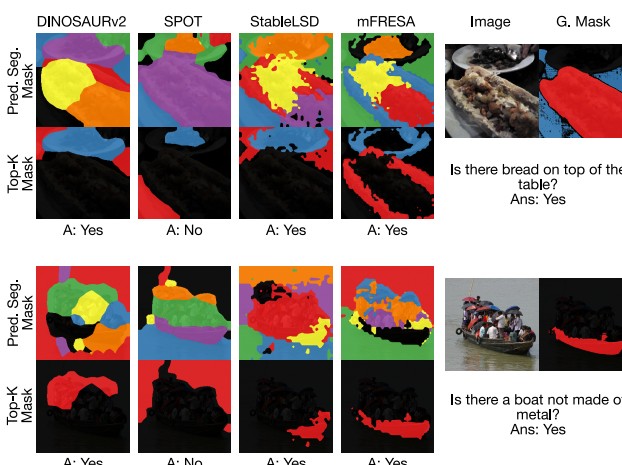

*Figure 5.* **Qualitative examples.** AwGA scores each example using the image, grounding masks (G. Mask), and the question–answer pair. The need for AwGA is evident: DINOSAURv2 achieves high G-Acc (correct ans. and high mIoU) but low AwGA, since the Top-$K$ answer-attributed slots poorly overlap with the grounded mask. Also, StableLSD predicts the correct answer, but has a low AwGA due to weak grounding overlap.

*Takeaway 4.* Using only object discovery or semantic prediction performance metrics provides an incomplete evaluation of OCL models. AwGA complements these metrics by jointly measuring *what* and *where*, and penalizes both localization and representation fragmentation.

### 4.5. Ablations

**Effect of LLM and connector.** We next show that our VLM-based evaluation scheme is robust to both LLM and connector choices in terms of the AwGA metric. To evaluate this, we compute the Spearman rank correlation of AwGA scores between Phi-2 (3B) and Qwen2 (7B) on the enhanced grounded GQA dataset. Then, using Phi-2 as the LLM, we evaluate against two popular connector variants, 1-layer MLP (MLP×1) and 2-layer MLP (MLP×2), and again report Spearman correlations. As seen in Fig. 6, the rank correlations remain consistently strong (>0.70), showing that AwGA rankings are stable across small and large-sized

*Table 4.* **Object discovery (OD) and representational quality are uncorrelated.** FT-DINOSAUR scores highest on OD metrics ($mBO_i$, mIoU) but underperforms on various VQA tasks (all in %, ↑). All methods use DINOv2 as the backbone. The Spearman's rank correlation between accuracy (VQAv2) and mIoU for these models is $-0.2$, indicating a negative correlation.

| Dataset | VQAv2 | Nat. Bench | Sugar C. | COCO | |
|---|---|---|---|---|---|
| Metric | | accuracy | | mIoU | $mBO_i$ |
| DINOSAURv2 | 63.84 | 3.37 | 75.98 | 27.25 | 28.42 |
| FT-DINOSAUR | 60.97 | 2.89 | 70.94 | **34.52** | **36.08** |
| StableLSD | 62.06 | 3.00 | 72.92 | 24.52 | 25.72 |
| mFRESA *(ours)* | **65.75** | **4.11** | **77.17** | 30.60 | 32.17 |

*Table 5.* **Performance comparison of different models using G-Acc and AwGA metrics** (all in %, ↑). The mIoU and Acc. measures exhibit a weak Spearman correlation (Phi2: 0.35 and Qwen2: 0.50), indicating that either metric alone is a poor proxy for evaluating both the localization and the usefulness of the representation of OCL methods. G-Acc and AwGA jointly evaluate both; however, AwGA additionally penalizes representation fragmentation (also see Fig. 7).

| LLM | Phi2 | | | | Qwen2-7B | | |
|---|---|---|---|---|---|---|---|
| Metric | mIoU | Acc. | G-Acc. | AwGA | Acc. | G-Acc. | AwGA |
| DINOSAUR | 50.52 | 60.13 | 30.80 | 11.64 | 64.05 | 32.71 | 13.47 |
| DINOSAURv2 | 47.99 | 66.27 | 32.40 | 11.92 | 68.54 | 33.18 | 12.44 |
| VQDINO$_{MLP}$ | 48.50 | 65.55 | 32.27 | 12.13 | 68.18 | 34.13 | 13.47 |
| FT-DINOSAUR | 59.09 | 61.05 | 33.94 | 13.25 | 65.21 | 36.28 | 15.36 |
| SPOT | 53.76 | 64.08 | 38.45 | 12.81 | 68.32 | 41.45 | 15.14 |
| Slot Diffusion | 54.91 | 61.54 | 34.39 | 12.42 | 65.75 | 36.91 | 13.85 |
| StableLSD | 47.94 | 64.64 | 31.53 | 11.49 | 69.02 | 33.84 | 13.37 |
| mFRESA *(ours)* | 56.92 | 67.58 | 39.20 | 13.91 | 71.41 | 41.33 | 15.88 |

LLMs and different connectors, indicating the robustness of our evaluations. Additional robustness results are provided in Table 10.

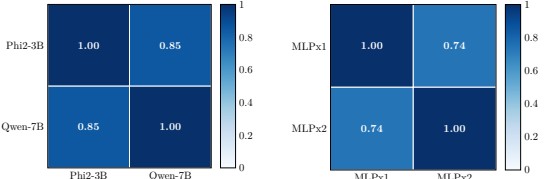

*Figure 6.* **Robustness of AwGA.** Spearman's rank correlation for the AwGA metrics for different LLM and connectors designs. AwGA remains stable across different LLMs and connector architectures, suggesting that our VLM-based evaluation is relatively insensitive to specific LLMs or connector architectures.

**Robustness of AwGA to the attribution method.** The AwGA metric requires selecting an attribution method as the *only* hyperparameter. In this experiment, we compare two common attribution methods: simply using the gradient information associated with each slot, and integrated gradients (Sundararajan et al., 2017). As seen in Table 6, the AwGA metric is very robust to the choice of attribution method (Spearman rank correlation of 0.77). We use gradient-based attribution for computational efficiency.

*Table 6.* **Attribution robustness.** The AwGA metric is robust to attribution choice, showing a strong Spearman correlation ($\rho = 0.77$) between gradient and integrated-gradient attributions.

| | Attr. Type | |
|---|---|---|
| | Grad. | Int. Grad. |
| DINOSAUR | 11.64 | 7.58 |
| DINOSAURv2 | 11.92 | 7.87 |
| FT-DINOSAUR | 12.81 | 8.26 |
| SlotDiffusion | 12.42 | 8.83 |
| StableLSD | 11.49 | 8.04 |
| mFRESA (ours) | **13.91** | **9.48** |

*Table 7.* **Importance of HOG and feature decoders.** Both HOG and (DINOv2) feature decoders improve localization performance (mIoU), downstream VQA performance, and joint evaluation metric, indicating their importance. All experiments in (%, ↑).

| Decoder Img. Feat. HOG | | | GQA | OOD | Sugar C. | VQAv2 | mIoU | AwGA |
|:---:|:---:|:---:|---|---|---|---|---|---|
| ✓ | – | – | 51.45 | 52.89 | 72.92 | 62.06 | 47.94 | 11.49 |
| ✓ | ✓ | – | 52.21 | 54.20 | 75.92 | 61.40 | 55.76 | 13.37 |
| ✓ | ✓ | ✓ | **53.90** | **55.27** | **77.27** | **65.58** | **56.92** | **13.91** |

**Choice of reconstruction objective.** mFRESA builds on StableLSD and includes two new decoders: the feature and HOG decoders. In Table 7, we quantify the effect of each decoder. We observe that simply adding the DINOv2 feature decoder improves results across almost all tasks (localization and representation usefulness). Additionally, incorporating HOG features further improves both abilities, as they provide object-edge information that helps capture object boundaries more accurately. This leads to better-localized slots and more informative slot representations, highlighting the effectiveness of both decoders in learning stronger object-centric representations. Overall, although mFRESA is computationally more expensive than StableLSD, it delivers consistent gains of at least 2 absolute percentage points across multiple benchmarks and metrics, including mIoU, accuracy, and AwGA.

**Limitations.** While our evaluation framework and AwGA provide a broader view of object-centric learning (OCL), they have some limitations. First, our protocol relies on training large vision-language models (VLMs) as evaluators, so our evaluation inherits their biases. Secondly, though our evaluation cost is amortized across many benchmarks, the one-time training for each OCL model remains expensive. Second, AwGA requires access to grounding masks for question–answer pairs, which currently limits its evaluation to our eGQA benchmark. Larger and more diverse grounded VQA datasets would help test whether our conclusions generalize beyond eGQA, but such datasets with question–answer-level box or mask annotations remain scarce. Nevertheless, we view AwGA as an important additional evaluation protocol for evaluating OCL models. Finally, our evaluation focused on slot-attention, which remain the most popular and widely adopted class of OCL models; we leave the evaluation of more recent adaptive slot-count methods (Fan et al., 2024; Liu et al., 2026) to future work.

## 5. Conclusion

Object-centric learning (OCL) has made notable progress in unsupervised object discovery (UOD) for real-world scenes. However, broader goals such as compositionality, counterfactual reasoning, and OOD robustness remain underexplored, partly because existing evaluation schemes require costly task-specific retraining. To address this, we propose a scalable evaluation protocol based on visual instruction tuning of an LLM. Our protocol evaluates how effectively a VLM can leverage slot representations from different OCL models across diverse VQA benchmarks, without training task-specific probes for each benchmark. We find that current OCL models, though competitive, still lag the DINOv2 encoder in absolute performance on several key benchmarks. This indicates the need for developing stronger OCL models. We then show the need for an evaluation metric that jointly evaluates the localization and representation usefulness of OCL models using a single unified metric. To this end, we introduce a grounded evaluation benchmark (eGQA) and propose Attribution-aware Grounded Accuracy (AwGA), a unified metric that jointly evaluates the "what" and "where" aspects of OCL. Finally, we include a simple multi-target reconstruction baseline (mFRESA) as a reference, demonstrating that combining reconstruction objectives improves reasoning and localization capabilities.

## Impact Statement

**Positive.** Object-centric learning (OCL) aims to represent scenes as sets of objects, which may support more structured and interpretable perception (Baillargeon et al., 1985; Téglás et al., 2011). This work contributes an evaluation framework that benchmarks the downstream utility of OCL representations across diverse reasoning-oriented VQA benchmarks (*e.g.*, compositionality, OOD generalization, and robustness) without training task-specific probes. We further identify fragmentation issues in existing evaluation protocols and propose eGQA, together with the AwGA metric, to jointly evaluate the *what* and *where* capabilities of OCL models. We hope these tools will support more reliable evaluation and accelerate progress toward OCL models that improve both the localization and the usefulness of learned object representations.

**Negative.** Our evaluation methodology for the OCL models is based on VLMs, which employ a pre-trained large language model (LLM). Thus, our evaluation can also inherit LLM biases; therefore, evaluating vision encoders with LLMs should be done cautiously. These biases can be eliminated by using and training VLM on a more balanced dataset. However, this is out of the scope of this work. Another concern with our evaluation framework is that training the VLM for evaluating the OCL models is not energy efficient, as standard metrics do not require such training. However, it should be noted that the amortized cost of evaluating OCL models within our evaluation framework is still lower than that of training separate linear or transformer-based probes to assess the capabilities of OCL methods.

**Acknowledgements.** This project has received funding from the European Research Council (ERC) under the European Union's

Horizon 2020 program (grant agreement No. 866008). The project was also supported in part by the Deutsche Forschungsgemeinschaft (DFG, German Research Foundation) under Germany's Excellence Strategy (EXC-3057/1 "Reasonable Artificial Intelligence", Project No. 533677015). SSM has been funded by the DFG – Project No. 529680848. We gratefully acknowledge support from the hessian.AI Service Center (funded by the Federal Ministry of Research, Technology and Space, BMFTR, grant no. 16IS22091) and the hessian.AI Innovation Lab (funded by the Hessian Ministry for Digital Strategy and Innovation, grant no. S-DIW04/0013/003)."

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

# A. mFRESA: Training details for mFRESA

mFRESA is trained on a single NVIDIA A100 GPU with 80GB of VRAM. The encoder and image decoder components closely follow the StableLSD setup (Jiang et al., 2023), with mFRESA introducing two additional modules: a HOG feature extractor (Dalal & Triggs, 2005; Wei et al., 2022) and decoder, as well as a DINOv2 feature decoder. The model is trained for 500K iterations on the COCO dataset (Lin et al., 2014). Images fed to the DINOv2 encoder (Oquab et al., 2024) are resized and center-cropped to 518×518 pixels. We used an Adam optimizer (Kingma & Ba, 2015) for training our model. The full training and architectural details of our method are shown in Table 8. The additional decoders also lead to additional training times, for example, StableLSD on a single A100 GPU trains in approx. 43 hours. Adding the feature decoder increases the training time to 65 hours. Adding the HOG decoder further increases the total training time to 82 hours.

*Table 8.* **Architectural and training details for mFRESA.**

| Module | Hyperparameter | Value |
|---|---|---|
| General | Batch size | 32 |
| | Precision | fp16 |
| | Learning rate | 2e-5 |
| | Learning rate scheduler | Constant |
| | Optimizer | Adam (Kingma & Ba, 2015) |
| | Adam ($\beta_1$, $\beta_2$) | (0.9, 0.999) |
| | Adam eps | 1e-8 |
| | Weight decay | 1e-2 |
| | Learning rate scheduler | Constant |
| | Iterations | 500K |
| | Max. grad norm | 1.00 |
| Encoder | Architecture | DINOv2 (Oquab et al., 2024) |
| | Patch size | 14 |
| | Backbone | ViT-B (Dosovitskiy et al., 2021) |
| | Embedding dimensions | 768 |
| Slot Attention | # Iterations | 5 |
| | # Slots | 7 |
| | Slot Size | 768 |
| Image Decoder | Architecture | Stable Diffusion (Rombach et al., 2022) |
| | Model version | 2.1 |
| Feat. Decoder | Architecture | MLP |
| | No. of layers | 2 |
| | Hidden dimensions | 1536 |
| HOG Decoder | Architecture | MLP |
| | No. of layers | 2 |
| | Hidden dimensions | 1536 |

# B. Additional results

## B.1. Additional application of our evaluation framework

**Quantifying the type of learned slots.** Our evaluation framework can be used to to probe whether architectural choices bias slots toward encoding specific properties (*e.g*., spatial, relational, or object). The GQA dataset (Hudson & Manning, 2019) categorises questions into four semantic types: *(1)* object (existence), *(2)* attribute (properties or position), *(3)* category (class membership), and *(4)* relation (subject–object relations). As shown in Table 9, feature reconstruction methods excel at existence and relation questions, whereas image-only methods like Slot Diffusion (Wu et al., 2023) and StableLSD (Jiang et al., 2023) lag behind. mFRESA, which combines both, achieves the best results in three of four categories. For MM-Vet (Yu et al., 2024), covering recognition and spatial queries, results are mixed: feature- and image-based approaches perform similarly on recognition, while Slot Diffusion performs best on spatial relations.

**Correlation analysis.** We evaluate the robustness of our slot-attention evaluation framework to the choice of the large language model (LLM). While the choice of LLMs affects the absolute performance of the models, we find that the relative ranking of OCL models remains largely unchanged across different LLMs. Table 10 reports Spearman's rank correlations between results obtained with Phi2 and Qwen2-7B across multiple datasets. The strong correlations ($\rho \geq 0.89$) indicate that our proposed VLM-based evaluation framework is stable, with model rankings preserved across LLMs.

**Qualitative results.** Fig. 7 presents qualitative examples comparing the predicted masks with attribution-aware masks obtained by selecting the slots with the $K$-highest attributions based on a gradient-based attribution method (Simonyan et al.,

*Table 9.* **Type of properties encoded by slots**. VLM-based evaluation (accuracy in %, ↑) allows us to quantify the type of properties that a slot encodes via a categorization of the questions.

| Dataset | GQA (Hudson & Manning, 2019) | | | | MM-Vet (Yu et al., 2024) | |
|---|---|---|---|---|---|---|
| | Attribute | Category | Object | Relation | Recognition | Spatial |
| DINOSAURv2 (Seitzer et al., 2023) | 57.58 | 45.26 | 78.02 | 46.95 | 21.5 | 23.9 |
| FT DINOSAUR (Didolkar et al., 2025) | 56.77 | 43.17 | **79.95** | 45.54 | 18.7 | 19.9 |
| SPOT (Kakogeorgiou et al., 2024) | 57.15 | 42.47 | 75.06 | 43.24 | 19.7 | 23.1 |
| Slot Diffusion (Wu et al., 2023) | 57.40 | 39.77 | 73.52 | 41.28 | 20.5 | **28.9** |
| StableLSD (Jiang et al., 2023) | 56.73 | 43.43 | 75.19 | 44.20 | 21.6 | 22.3 |
| mFRESA *(ours)* | **59.08** | **46.74** | 77.63 | **47.23** | **21.9** | 22.1 |

*Table 10.* **Spearman rank correlations between different models when using Phi2 and Qwen2-7B models as LLMs.** The results show that our evaluation framework is robust to the choice of LLM, and the rank between the models remains largely preserved (very strong correlation). Spearman $\rho$ (max 1, ↑).

| Dataset | GQA | POPE | MME | MMVet | VQAv2 | OOD | Nat. Bench | Sugar C. | AwGA |
|---|---|---|---|---|---|---|---|---|---|
| Spearman $\rho$ | 0.98 | 0.95 | 0.70 | 0.76 | 0.98 | 0.86 | 0.91 | 0.85 | 0.92 |

2014). These results highlight that high-quality predicted masks or accuracy alone do not always imply the usefulness of the slot representations. For example, in *top row*, StableLSD, despite having a good predicted mask, produces an incorrect output. Importantly, in *third row*, SPOT and StableLSD produce a correct response to the question, but use the wrong slot (as seen by the Top-k mask) to answer the question.

**Mean and standard deviation of results.** Training the VLMs with different random seeds and evaluating the resulting models is computationally intensive, as these VLMs are trained across 8 NVIDIA A100 GPUs, with the fine-tuning stage typically taking around 24 hours. This makes it infeasible to provide results from multiple runs using this approach. Instead, we report mean and standard deviation results for mFRESA and several baseline models on representative datasets during *evaluation*. We set the temperature for LLM generation to 0.02 and averaged the results across five random seeds (42, 1337, 2025, 4378, 8921). We report the results on SugarCrepe (Hsieh et al., 2023), MME (Fu et al., 2025), and POPE (Li et al., 2023b) as representative datasets for visual question answering in Table 11. We use the accuracy as an evaluation metric for the SugarCrepe and POPE datasets. For the MME dataset, we provide scores based on the MME evaluation script (with 2000 as the maximum for the perception task). Please note that the numbers reported in Table 2 and Table 3 of the main paper are for a temperature value set to 0. Comparing these to Table 11, we observe the ranking of the models following the same trend as with temperature 0. Setting the temperature $> 0$ introduces randomness into the output of large language models, enabling us to obtain the mean and standard deviation during evaluation. We observe small performance variation and consistent model rankings across random seeds, indicating the robustness of our VLM-based evaluation framework.

*Table 11.* **Mean and standard deviation of results.** Performance comparison with mean and standard deviation of different methods in a selection of representative datasets. SugarCrepe (Hsieh et al., 2023) and POPE (Li et al., 2023b) are evaluated in terms of accuracy (in %, ↑), MME (Fu et al., 2025) in terms of its score (↑). We observe small performance variation and consistent model rankings across random seeds, indicating the robustness of our VLM-based evaluation framework. We highlight the **best** and second best model among slot-attention methods.

| Method | SugarCrepe (Hsieh et al., 2023) | MME (Fu et al., 2025) | POPE (Li et al., 2023b) |
|---|---|---|---|
| DINOv2 | 82.14 ± 0.13 | 1283.96 ± 07.29 | 82.08 ± 0.10 |
| DINOSAURv2 | 76.20 ± 0.25 | 1123.47 ± 12.63 | 81.84 ± 0.24 |
| FT-DINOSAUR | 71.25 ± 0.23 | 1016.15 ± 22.11 | 81.54 ± 0.18 |
| SPOT | 71.65 ± 0.15 | 1066.04 ± 07.19 | 79.69 ± 0.04 |
| Slot Diffusion | 70.23 ± 0.33 | 1090.75 ± 08.09 | 79.74 ± 0.12 |
| StableLSD | 72.89 ± 0.32 | 1126.06 ± 17.21 | 81.13 ± 0.11 |
| mFRESA *(ours)* | **77.18 ± 0.27** | **1184.40 ± 19.52** | **82.20 ± 0.08** |

**Comparison to DINOv3.** We also compare our results to modern self-supervised learning methods such as DINOv3 (Siméoni et al., 2026). As shown in Table 12, DINOv3 improves over DINOv2 (Oquab et al., 2024) on benchmarks such as GQA, OODCV, MMVet, and POPE. The gains are generally modest and not consistent across all metrics. In particular, DINOv3 performs slightly worse on VQAv2, SugarCrepe, CVQA, and MME. Overall, DINOv3 raises the SSL upper bound only marginally, and the gap between OCL methods and SSL-based models widens only slightly on most VQA benchmarks.

*Table 12.* Comparison between DINOv2 and DINOv3 across VQA and compositional reasoning benchmarks. DINOv3 improves performance on some benchmarks, but the gains are modest and not consistent across all metrics.

| Model | GQA | VQAv2 | OODCV | SugarCrepe | CVQA | | MME | MMVet | POPE |
|---|---|---|---|---|---|---|---|---|---|
| | | | | | Direct | Boolean | | | |
| DINOv2 | 57.77 | 71.15 | 58.00 | 82.05 | 36.96 | 63.72 | 1279.21 | 22.6 | 82.01 |
| DINOv3 | 58.84 | 70.43 | 62.13 | 81.64 | 36.78 | 63.45 | 1264.17 | 23.7 | 84.45 |
| mFRESA | 53.90 | 65.58 | 55.57 | 77.27 | 38.09 | 66.64 | 1187.05 | 18.5 | 82.12 |

## C. Datasets

Here we describe the datasets used in Sec. 4.1 for our VQA-based evaluation of OCL models.

**VQAv2.0** (Goyal et al., 2017) is a dataset of 265,016 images from COCO and abstract scenes, each paired with an average of 5.4 open-ended questions requiring vision, language, and commonsense reasoning. Each question includes 10 ground-truth answers and 3 plausible but likely incorrect answers, making it a robust benchmark for evaluating visual question-answering (VQA) models.

**GQA** (Hudson & Manning, 2019) is a VQA dataset for real-world images that requires visual, spatial, and compositional reasoning. Importantly, GQA provides grounding masks (referred objects for answering questions) for each question in the validation set.

**POPE** (Li et al., 2023b). The Polling-based Object Probing Evaluation (POPE) assesses object-level perception and hallucination in vision-language models by querying whether specific objects are present in images. It consists of three settings: *(i)* Random – this setting samples absent objects at random, *(ii)* Popular – this setting selects missing objects from a frequently occurring object pool, and *(iii)* Adversarial – this setting targets commonly co-occurring but visually absent objects to challenge the model's grounding ability. In total, POPE consists of 3 sets of image-question pairs, each containing 1500 pairs with answer "Yes" and 1500 pairs with answer "No".

**MME** (Fu et al., 2025) is a comprehensive benchmark designed to evaluate the capabilities of multimodal large language models (MLLMs) across 14 diverse subtasks spanning both perception and cognition. In our work, we focus specifically on perception tasks, including coarse-grained recognition (existence, count, position, color), fine-grained recognition (poster, celebrity, scene, landmark, artwork), and optical character recognition (OCR). Model performance on these tasks is measured using the perception score, capped at 2000 points.

**MM-Vet** (Yu et al., 2024). Unlike standard evaluation benchmarks, MM-Vet evaluates the integration of key vision-language (VL) capabilities, such as recognition, optical character recognition (OCR), knowledge reasoning, language generation, spatial understanding, and mathematical reasoning. MM-Vet contains 200 images and 218 questions, each paired with its respective ground truth.

## D. Sensitivity analysis

**Sensitivity to temperature.** We evaluate the sensitivity of AwGA to the VLM's decoding temperature. As shown in Table 13 *(top-left)*, AwGA is stable for temperatures between 0.0 and 0.2, with only small score variations across models. At a higher temperature of 0.5, performance mildly decreases for all methods, suggesting that more stochastic decoding can make the attribution-based evaluation slightly noisier. Importantly, the relative rankings of methods remain highly consistent across temperatures: mFRESA performs best across all settings, while the remaining models follow a similar ordering.

**Sensitivity to prompt templates.** We further test whether AwGA is sensitive to the exact wording of the evaluation prompt. We compare the original LLaVA short-answer prompt with two simple prompt variants, Prompt 1: "<Question?> Provide a short answer in one word or phrase.", Prompt 2: "<Question?> Answer in one word if possible; otherwise use a short phrase.". As shown in Table 13 *(top-right)*, AwGA scores remain very similar across all prompt templates, and the relative ranking of the compared models is unchanged. These results suggest that AwGA is robust to minor changes in prompt wording and does not depend heavily on a specific prompt template.

**Sensitivity to connector choice.** We also evaluate the sensitivity of AwGA to the connector architecture used to map visual tokens to the language model. We already reported results with 1-layer and 2-layer MLP connectors in the main paper. We additionally evaluate the robustness to a QFormer connector (Li et al., 2023a). As shown in Table 13 *(bottom-left)*, QFormer

*Table 13.* **Sensitivity analysis of AwGA.** *Top-left*: sensitivity to decoding temperature. *Top-right*: sensitivity to prompt templates. *Bottom-left*: sensitivity to connector choice. *Bottom-right*: sensitivity to eGQA filtering. AwGA remains stable across temperatures 0.0–0.2 and across simple prompt variants, with consistent model rankings. For connector choice, QFormer gives lower absolute scores than the MLP connector, but model rankings remain strongly correlated. Without eGQA filtering, absolute scores decrease, but the relative ordering is largely preserved.

| Model | 0.0 | 0.1 | 0.2 | 0.5 |
|---|---|---|---|---|
| DINOSAUR | 11.64 | 11.78 | 12.07 | 11.07 |
| SPOT | 12.81 | 12.62 | 12.56 | 12.04 |
| SlotDiffusion | 12.42 | 12.70 | 12.55 | 11.99 |
| StableLSD | 11.49 | 11.51 | 11.51 | 10.79 |
| mFRESA | **13.91** | **13.98** | **13.94** | **13.43** |

| Model | Std. | Prompt 1 | Promp 2 |
|---|---|---|---|
| DINOSAUR | 11.64 | 11.83 | 11.69 |
| SPOT | 12.81 | 12.67 | 12.63 |
| SlotDiffusion | 12.42 | 12.62 | 12.44 |
| StableLSD | 11.49 | 11.53 | 11.14 |
| mFRESA | **13.91** | **14.16** | **13.98** |

| Model | MLP2x | QFormer |
|---|---|---|
| DINOSAURv2 | 11.92 | 4.65 |
| SPOT | 12.81 | 5.11 |
| SlotDiffusion | 12.42 | 5.31 |
| mFRESA | **13.91** | **8.04** |

| Model | Filtered | Unfiltered |
|---|---|---|
| DINOSAUR | 11.64 | 6.29 |
| SPOT | 12.81 | 6.71 |
| SlotDiffusion | 12.42 | **6.94** |
| StableLSD | 11.49 | 6.00 |
| mFRESA | **13.91** | 6.71 |

leads to lower absolute AwGA scores than the 2-layer MLP connector. However, the rankings remain strongly correlated with the MLP connector, with Pearson and Spearman correlations of at least $0.8$. This suggests that AwGA is reasonably robust to the choice of connector architecture, even when absolute scores differ.

**Sensitivity to eGQA filtering.** We also report results for the representative models used in the main comparison without applying filtering based on the number of objects or mask coverage in eGQA. For this unfiltered setting, we cap the evaluation to 20K samples for efficiency. As seen in Table 13 *(bottom-right)*, the absolute AwGA scores decrease without filtering, the relative ordering is largely preserved, with Pearson's $\rho = 0.7$, suggesting that our conclusions are robust to the chosen filtering thresholds.

# E. Enhanced GQA dataset

We construct our enhanced GQA (eGQA) dataset used in Sec. 4.4 based on the validation split of the original GQA dataset (Hudson & Manning, 2019). Our enhanced version comprises 10,000 questions, each accompanied by grounded segmentation masks. To convert grounding bounding boxes—*i.e.*, the coordinates of objects referenced in the questions—into segmentation masks, we utilize the SAM2 model (Ravi et al., 2025), specifically the "sam2.1-heira-large" checkpoint with its default configuration.

To ensure relevance and clarity, we apply filtering criteria that discard images with more than 7 bounding boxes or with total box coverage less than 10% of the image area. These thresholds are chosen to retain only prominent objects while maintaining compatibility with object-centric learning (OCL) models trained on the COCO dataset (Lin et al., 2014), which typically utilize seven slots. Additional examples from our eGQA dataset are shown in Fig. 8.

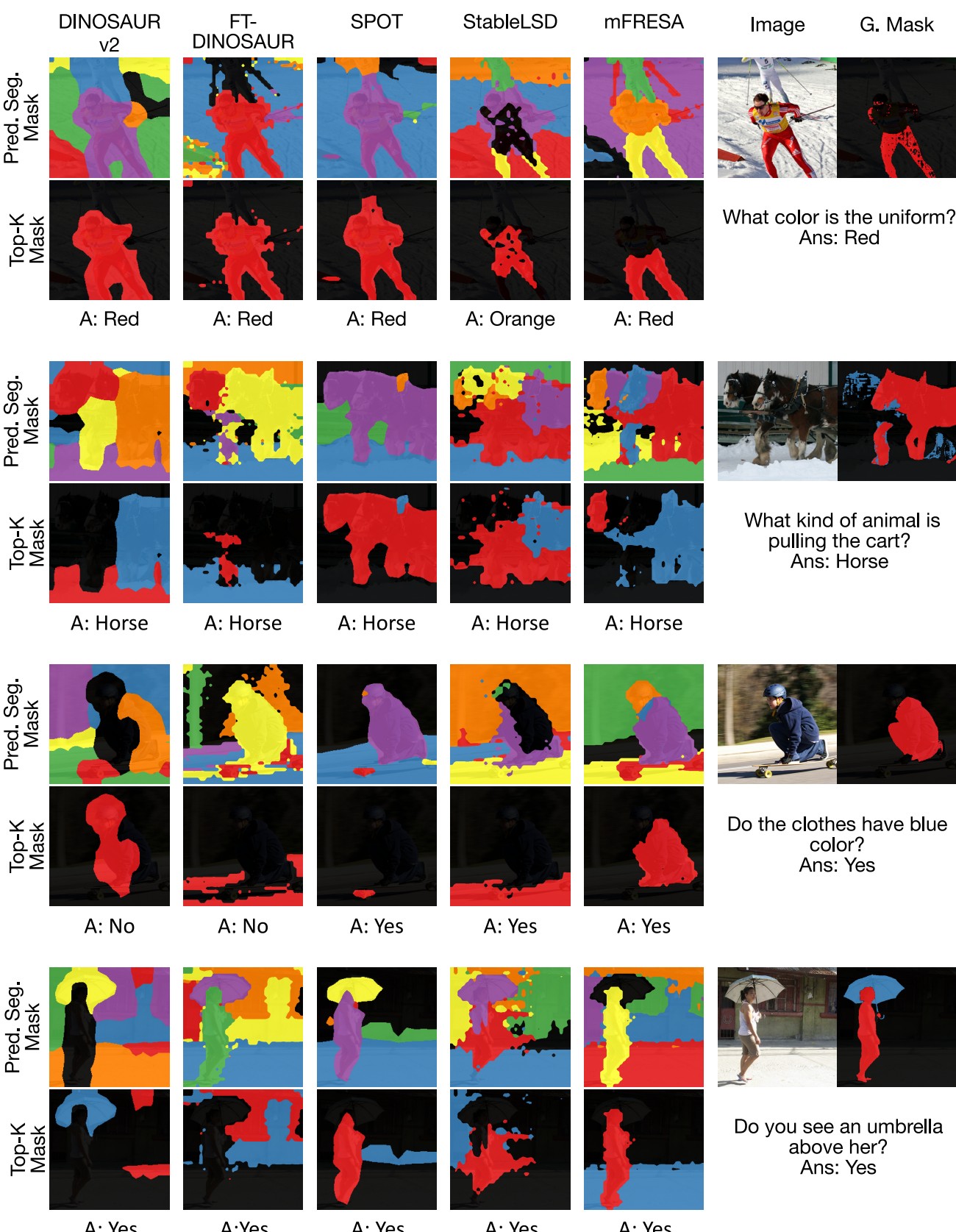

*Figure 7.* **Qualitative examples.** We visualize predicted masks and attribution-aware masks obtained by selecting the $K$ tokens with the highest attributions. A good predicted mask does not necessarily imply a correct slot encoding, which can lead to wrong answers (See text). G. Mask denotes the ground-truth eGQA mask for the regions required to answer each question. Pred. Seg. Mask denotes the predicted mask from the slot attention module.

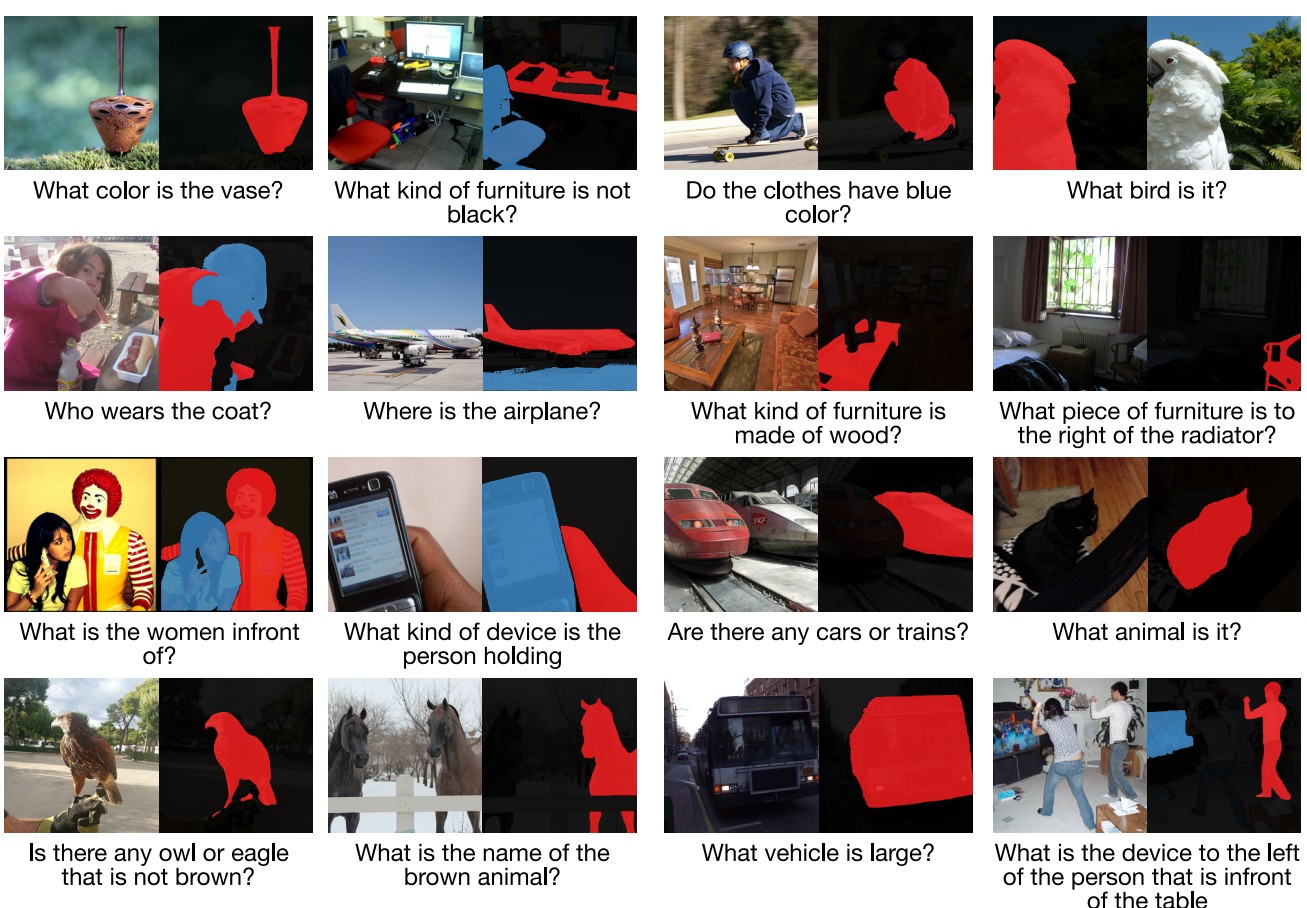

*Figure 8.* **Additional samples from the enhanced GQA dataset.** The dataset is composed of questions and answers (not shown) pairs. The grounding masks denote the masks of the referring objects required to answer the question.

