# OpenReview forum: "Evaluating Object-Centric Models beyond Object Discovery"
_ICML.cc/2026/Conference — ICML 2026 regular_

### Official Review · Reviewer_Dv7p · 2026-03-06

**Soundness:** 2
**Presentation:** 3
**Significance:** 3
**Originality:** 4
**Overall Recommendation:** 5
**Confidence:** 4

**Summary:**

This paper proposes a broader evaluation framework for object-centric learning by combining VLM-based probing across diverse VQA tasks with a new joint grounding metric (AwGA) on an enhanced grounded dataset (eGQA). The core message is that standard object-discovery metrics alone are weak proxies for downstream reasoning utility, and that evaluating "what" and "where" jointly gives a more realistic picture of representation quality. Results suggest OCL methods can be competitive with far fewer tokens than dense encoders, while still exposing clear gaps on harder compositional and adversarial benchmarks.

**Compliance With Llm Reviewing Policy:**

Affirmed.

**Final Justification:**

My main concerns have been adequately addressed in the Rebuttal. The authors clarified the amortized-cost claim, provided the requested parameter/training/inference comparisons, addressed the DINOv3 question, and committed to improving both the presentation and related-work discussion. Overall, the rebuttal resolves my main concerns sufficiently, and I therefore raise my score from 4 to 5 (Accept).

**Key Questions For Authors:**

1. What are the training and inference compute requirements (FLOPs, wall-clock time, GPU-hours) of the compared methods, and how do they compare in parameter count?
2. What are the parameter, token, and compute trade-offs between the OCL encoders and the foundation-model baselines?
3. Would using DINOv3 change your conclusion?

**Limitations:**

yes

**Strengths And Weaknesses:**

## Strengths
- Interesting empirical result: object discovery metrics and downstream representational utility appear weakly/negatively correlated (Table 4), which is important for the OCL community.
- Strong practical result: several OCL methods are competitive on VQA while using far fewer visual tokens than DINOv2 (Table 2), and some OCL variants perform well on specific robustness tasks (Table 3).
- The unified "what + where" evaluation framing (AwGA + eGQA) is a useful direction and makes a clear argument against disjoint evaluation.

## Weaknesses
- Figure 1 is hard to parse: slot fragmentation and attribution are not visually obvious enough (color choices and arrow placement reduce readability). This is the first figure and should communicate the failure modes immediately.
- Missing related-work citation: *Compositional Scene Representation Learning via Reconstruction: A Survey* [1] should be discussed in `Related Work`.
- The scalability/no-retraining claim is currently overstated: the method still trains the connector and (in Stage II) the full LLM, which can be expensive. Please explain this on more detail. The Amort.
cost from Table 1 is to unspecific.
- mFRESA is central to the empirical argument but under-explained in the main paper. Even if not a core contribution, it needs a compact architecture figure or a clearer main-text description.
- Efficiency significance is hard to judge: while token count is mentioned (7 vs 196), there is no direct table/plot comparing parameter counts and training/inference cost versus large foundation encoders.
- Baseline coverage is dated: only DINOv2 is used as the main foundation-model comparator; DINOv3 [2] (released in Aug 2025) should be included or explicitly discussed as a limitation.
- Given the paper's "what + where" framing, please discuss prior OCL work that explicitly separates appearance and location (e.g., Loci [3])

---
[1] Jinyang Yuan, Tonglin Chen, Bin Li, and Xiangyang Xue. *Compositional Scene Representation Learning via Reconstruction: A Survey*. arXiv:2202.07135, 2022 (revised 2023). https://arxiv.org/abs/2202.07135

[2] Oriane Simeoni, Huy V. Vo, Maximilian Seitzer, Federico Baldassarre, Maxime Oquab, et al. *DINOv3*. arXiv:2508.10104, 2025. https://arxiv.org/abs/2508.10104

[3] Manuel Traub, Sebastian Otte, Tobias Menge, Matthias Karlbauer, Jannik Thuemmel, and Martin V. Butz. *Learning What and Where: Disentangling Location and Identity Tracking Without Supervision*. arXiv:2205.13349, 2022 (revised 2023). https://arxiv.org/abs/2205.13349

---

> ### Author Rebuttal · Authors · 2026-03-31
>
> We thank the reviewer for the thoughtful review, positive assessment, and constructive suggestions for improving the paper. Below, we address each concern in turn.
>
> **W2, W7 Additional related work**: We thank the reviewer for these references and will add them to the related work section. We focused our empirical comparison on OCL methods that are designed for real-world images, which is why we did not include Loci in the benchmark; we will clarify this scope explicitly and discuss its relevance to our “what + where” framing.
>
> **W3 Scalability overstated claims**: We would like to clarify that our claim concerns a lower amortized benchmarking cost, not a zero training cost. Our VLM-based probe is trained once and then reused across many evaluation dimensions without task-specific retraining. For mFRESA, this single VLM training takes about 24 hours and supports 8 benchmarks in our paper, corresponding to a much lower amortized benchmark cost than task-specific retraining, whose cost scales with the number of benchmarks. This amortized cost decreases further as more benchmarks are added. By contrast, conventional probing requires separate retraining and tuning for each task/dataset, even for simple probes, and therefore scales linearly with the number of benchmarks.
>
> **W1, W4 Presentation improvements**: We thank the reviewer for this comment. Due to space considerations, we decided to move mFRESA to the supplemental section. We will add a concise main-text description of mFRESA, including its reconstruction targets and why they improve object-centric representations, to the main paper for the camera-ready. Additionally, we will use more distinct colors for Fig. 1 and improve its overall readability.
>
> **W5, KQ1&2 Additional details of parameters and inference times**: In our work, we do not claim OCL methods to be more efficient than SSL methods (DINOv2). The comparison to DINOv2 is only meant to give an illustrative SSL upper bound in performance (L220). We did want to point out that, despite using far fewer tokens, the performance gap between OCL methods and DINOv2 is not that large. However, on key dimensions such as OOD generalization and adversarial robustness, settings where object-centric methods are often hypothesized to help most, we find that OCL representations still lag behind SSL representations. We now present the parameter counts, inference, and training times for each model in our work.
>
> | Model | Parameters | Avg. inference time with VLM | Training time |
> |--|--|--|--|
> | DINOv2 | 86.5M | 2.99s | 3.3 days (96 x A100GPUs)* |
> | DINOSAUR | 98.25M | 3.90s | 44h (1x A6000 Ada) |
> | DINOSAURv2 | 99.83M | 3.89s | – |
> | FT-DINOSAUR |100.12M | 3.69s | 2-3 days* (1x A100 GPU) |
> | SPOT | 212.9M | 4.28s | 48h* (1x A5000 GPU) |
> | SlotDiffusion | 171.75M | 3.82s | 40h* |
> | StableLSD | 95.06M + 865.9M (StableLSD decoder frozen) |  3.60s | 43h (1x A100 GPU) |
> | mFRESA |102.53M + 865.9M (StableLSD decoder frozen) | 3.62s | 82h (1x A100 GPU) |
>
> These numbers support our main claim: OCL encoders use far fewer visual tokens than dense SSL encoders and remain competitive on several tasks.  * denotes the training times are taken from the papers or the official GitHub repos of the methods.
>
> **W6, KQ3 Additional results**: As noted earlier, we use DINOv2 only as an illustrative upper-bound baseline. Adding DINOv3, therefore, does not change the main conclusion of our paper. While DINOv3 improves the upper-bound performance, the gains are modest, and the gap between OCL methods and SSL-based models widens only slightly on most VQA benchmarks.
>
> Model | GQA | VQAv2 | OOD | Sugar Crepe | CVQA(Direct) | CVQA (Boolean) | MME | MMVet| POPE |
> | -- | -- | -- | -- | -- | --| -- | -- | -- | -- |
> | DINOv2 | 57.77 | 71.15 | 58.00 | 82.05 | 36.96 | 63.72 | 1279.21 | 22.6 | 82.01 |
> | DINOv3 | 58.84 | 70.43 | 62.13 | 81.64 | 36.78 | 63.45 | 1264.17 | 23.7 | 84.45 |

---

> > ### Author Rebuttal · Reviewer_Dv7p · 2026-04-01
> >
> > My concerns have been adequately addressed. The authors clarified the amortized-cost claim, provided the requested parameter/training/inference comparisons, addressed the DINOv3 question, and committed to improving both the presentation and related-work discussion. Overall, the rebuttal resolves my main concerns sufficiently, and I therefore raise my score from 4 to 5 (Accept).

---

> > > ### Author Response · Authors · 2026-04-04
> > >
> > > Thank you for recognizing that the concerns have been fully resolved and for updating your assessment accordingly. We appreciate your constructive feedback, which helped strengthen our manuscript.

---

### Official Review · Reviewer_v2Dz · 2026-03-10

**Soundness:** 3
**Presentation:** 3
**Significance:** 3
**Originality:** 3
**Overall Recommendation:** 5
**Confidence:** 5

**Summary:**

This work argues that common evaluation protocols for object-centric learning (OCL), especially unsupervised object discovery, do not adequately reflect whether slot representations encode useful object properties for downstream reasoning, robustness, and compositional generalization. The authors propose a VLM-based evaluation protocol (LLaVA-style visual instruction tuning) where OCL slots serve as visual tokens, enabling broad benchmarking across multiple VQA-style datasets without training separate task-specific probes.
To address inconsistencies from evaluating “what” and “where” separately, the paper introduces Attribution-aware Grounded Accuracy (AwGA) and constructs an enhanced grounded GQA benchmark (eGQA) by converting GQA boxes into masks for attribution-aware evaluation.
Finally, the authors propose mFRESA, a simple multi-target reconstruction baseline (pixels + DINOv2 features + HOG) and report improved results over prior slot-based methods across several tasks.

**Compliance With Llm Reviewing Policy:**

Affirmed.

**Final Justification:**

My main concerns were adequately addressed in the rebuttal. The additional clarifications and experiments strengthened the paper and made the empirical conclusions more convincing. Overall, the rebuttal improved my assessment of the work, and I therefore raised my score from 4 to 5.

**Key Questions For Authors:**

1. Positioning / novelty: Can the authors more explicitly explain which of the main empirical conclusions would not be recoverable with prior probing approaches (e.g., transformer probes trained on VQA) and why? More concretely, what new conclusions arise from (i) the LLaVA-style instruction tuning, and (ii) AwGA?
2. The paper finds OCL methods lag behind DINOv2 on SugarCrepe, whereas some OC works such as [1] report advantages in (often synthetic) compositional OOD settings. Could you provide a more detailed explanation of what differs here (dataset biases, language priors, nature of compositional shifts, or limitations of slot representations)?
3. AwGA robustness beyond the current checks: You show stability across two attribution methods and some connector/LLM choices. How sensitive are AwGA rankings to (i) prompt templates, (ii) decoding parameters (temperature/beam), and (iii) small changes to connector capacity/training length? Even a small sensitivity study would increase confidence in AwGA as a benchmark metric.
4. How do the SAM2 conversion and filtering thresholds affect results? For example, do model rankings change if you vary (i) the “>7 boxes” threshold, or (i) the coverage threshold? This seems particularly important because the evaluation is meant to judge object-centric representations, yet the dataset is shaped by slot-count assumptions.
5. I would appreciate it if the authors could elaborate on whether the paper’s conclusions change if methods are evaluated with different slot counts at inference.

Overall, I believe this paper is taking an important and necessary step towards better understanding the role of object-centric learning, and I would recommend the acceptance of the paper.


[1] Kapl, Ferdinand, et al. "Are Object-Centric Representations Better At Compositional Generalization?." _arXiv preprint arXiv:2602.16689_ (2026).

**Limitations:**

See the questions and weaknesses above.

**Strengths And Weaknesses:**

### Strengths

1. The paper is well written and easy to follow.
2. The VLM-based evaluation protocol is a practical way to scale evaluation across many benchmarks (perception + robustness + counterfactual/compositional) using a unified interface, and is an important evaluation protocol that was missing for OCL.

### Weaknesses

1. Novelty relative to prior work is still slightly underspecified. The core message (object discovery ≠ downstream performance; comparison to foundation models; correlation analyses) has appeared in previous works in different forms. The main clear novelties seem to be (i) the specific _visual-instruction-tuning_ evaluation pipeline for slots, (ii) AwGA (+ eGQA), and (iii) mFRESA. I think the paper would benefit from an even sharper framing of what new insight is enabled only by this pipeline/metric beyond what earlier evaluation setups could show.

---

> ### Author Rebuttal · Authors · 2026-03-31
>
> We thank the reviewer for helpful feedback and address each point below.
>
> **W1, KQ1 Novel insights from this work**:
> *(i)* Our LLaVA-style instruction tuning reuses the same evaluation mechanism across many benchmarks, enabling consistent comparison of OCL methods across compositionality, OOD robustness, counterfactual reasoning, and grounded reasoning. In contrast, training separate transformer probes is cumbersome, dataset-specific, and often infeasible on small diagnostic benchmarks (e.g., OODCV). To the best of our knowledge, no prior work has provided a unified evaluation of OCL methods across such diverse dimensions on real-world benchmarks under a shared protocol. Our framework reveals, for example, that object-centric representations, despite using far fewer tokens, match DINOv2 on hallucination reduction (POPE), but lag on adversarial corruptions (N.Bench). *(ii)* AwGA enables joint evaluation (‘what’ and ‘where’) of OCL methods. Disjoint metrics can make model comparison ambiguous (e.g., one model best on mIoU, another on accuracy). AwGA combines both aspects into a single score and exposes failure modes missed by prior metrics, such as correct semantics with weak localization or correct answers from the wrong slot.
>
> **KQ2. Differences with [1]**: Two main differences explain the divergence between [1] and our work. (a) Unlike our work, which compares frozen OCL models against frozen DINOv2, [1] trains OCL models directly on CLEVRTex and compares them to a frozen DINOv2 baseline. Since DINOv2 is trained on natural images, the synthetic-natural domain gap hurts its performance in [1]. (b) [1] evaluates OCL methods on the compositional out-of-distribution (cOOD) split of the CLEVRTex dataset (unseen combinations within the same domain). This evaluation inherently mixes compositional and distributional shifts. In contrast, our benchmark suite evaluates these aspects separately (Table 2), allowing a clearer comparison between compositional and OOD behavior. Taken together, these differences explain the divergence between our results and those reported in [1].
>
> **KQ3: Sensitivity analysis**
>
> (i) **Decoding temperature**:
> We present results for the main compared OCL models at different decoding temperatures.
>
> | Model | 0.0 | 0.1 | 0.2 | 0.5 |
> |--|--|--|--|--|
> | DINOSAUR | 11.64 | 11.78 | 12.07 | 11.07 |
> | SPOT | 12.81 | 12.62 | 12.56 | 12.04 |
> | SlotDiffusion | 12.42 |  12.70 | 12.55 | 11.99 |
> | StableLSD | 11.49 | 11.51 | 11.51| 10.79 |
> | mFRESA | 13.91 | 13.98| 13.94 | 13.43 |
>
> As seen, AwGA is stable across temperatures 0.0–0.2, with only mild degradation at 0.5; method ranking remains highly stable across temperatures.
>
> (ii) **Prompt templates**: We tested two simple variants of the standard LLaVA short-answer prompt. Across all compared models, AwGA scores remain very similar, and the relative ranking is unchanged, showing robustness to simple prompt changes.
>
> | Model | Org. Prompt | Prompt 1 | Prompt 2 |
> |--|--|--|--|
> | DINOSAUR | 11.64 | 11.83 | 11.69 |
> | SPOT | 12.81 | 12.67 | 12.63 |
> | Slot Diffusion | 12.42 | 12.62 | 12.44 |
> | StableLSD | 11.49 | 11.53 | 11.14 |
> | mFRESA | 13.91 | 14.16 | 13.98 |
>
> (iii) **Connector choice**:  We already reported 1xMLP and 2xMLP results. They are trained for only 1 epoch, so reducing the training length is not suitable. We additionally tested another connector architecture (QFormer) on a subset of models. Although absolute AwGA scores are lower, rankings remain strongly correlated with the MLP connector (Pearson/Spearman ≥ 0.8), suggesting reasonable robustness to connector choice.
>
> | Model | MLP2x | QFormer |
> |--|--|--|
> | DINOSAURv2 | 11.92 | 4.65 |
> | SPOT | 12.81 | 5.11 |
> | SlotDiffusion | 12.42 | 5.31 |
> | mFRESA | 13.91| 8.04 |
>
> **KQ4: Threshold effect on eGQA construction**: We report results for the representative models used in the main comparison, with no filtering applied to the number of objects or coverage in the eGQA dataset. For this unfiltered setting, we capped the evaluation to 20K samples for efficiency.
> | Model | eGQA (filtered) | eGQA (unfiltered) |
> |--|--|--|
> | DINOSAUR | 11.64 | 6.29 |
> | SPOT | 12.81 | 6.71 |
> | Slot Diffusion | 12.42 | 6.94 |
> | StableLSD | 11.49 | 6.00 |
> | mFRESA | 13.91 | 6.71 |
>
> Although absolute AwGA scores decrease without filtering, the relative ordering is largely preserved (Pearson's rho~=0.7), suggesting that our conclusions are robust to filtering thresholds.
>
> **KQ5. Changing slot count at inference**: Slot count is an important inference-time hyperparameter and can affect all OCL metrics [2], including AwGA. To ensure fair comparison, as in prior OCL work, we use each method’s standard pretrained slot configuration, avoiding conflating representation quality with inference-time retuning. We will clarify this limitation in the revision.
>
> [2] Zimmerman et al., Sensitivity of Slot-Based Object-Centric Models to their Number of Slots, arXiv, 2023.

---

> > ### Author Rebuttal · Reviewer_v2Dz · 2026-04-02
> >
> > Thank you for the detailed rebuttal. I appreciate the additional clarifications and experiments, which addressed my main concerns and strengthened the paper. Overall, the rebuttal increased my confidence in the work, and I have therefore raised my score accordingly.

---

> > > ### Author Response · Authors · 2026-04-04
> > >
> > > Thank you for recognizing that the concerns have been fully resolved and for updating your assessment accordingly. We appreciate your constructive feedback, which helped strengthen our manuscript.

---

### Official Review · Reviewer_WLYs · 2026-03-12

**Soundness:** 3
**Presentation:** 3
**Significance:** 3
**Originality:** 3
**Overall Recommendation:** 5
**Confidence:** 4

**Summary:**

This paper addresses the evaluation of object-centric learning (OCL) models, noting that existing methods—unsupervised object discovery or linear probing—either poorly reflect downstream utility or require costly retraining and cannot jointly assess localization and representation quality. To address this, the authors propose a novel evaluation framework that (1) leverages vision-instruct-tuned LLMs as VLMs with OCL models as visual encoders for zero-shot evaluation on multiple VQA benchmarks, and (2) introduces Attribution-aware Grounded Accuracy (AwGA), a unified metric combining answer correctness, mask overlap, and gradient-based slot attribution to jointly measure localization and representation quality while penalizing fragmentation. Extensive experiments across multiple OCL models and VQA datasets demonstrate the stability and effectiveness of the framework, revealing strengths and weaknesses of current OCL approaches.

**Compliance With Llm Reviewing Policy:**

Affirmed.

**Final Justification:**

The rebuttal has resolved my concerns thus I would like recomend to accept this paper.

**Key Questions For Authors:**

The appendix shows that mFRESA takes 82 hours to train, almost double StableLSD’s 43 hours. Do you think the performance gains in Table 7 justify this extra cost? If resources were limited, which setup would you lean towards?

**Limitations:**

Yes. The authors openly discuss the limitations in the “Impact Statement” section: their VLM-based evaluation can inherit biases from the underlying LLM, and training these VLMs isn’t exactly energy-efficient.

**Strengths And Weaknesses:**

Strengths
1.The idea of using instruction-tuned VLMs as zero-shot probes for OCL representations is clever. It avoids the high cost that usually comes with traditional probing methods.
2.The AwGA metric links semantic answers with their corresponding spatial slots, which better reflects the model’s real performance than reporting accuracy and mIoU separately.
3.The experimental setup is solid. It covers a range of OCL models, different LLM backbones, connector designs, and several VQA benchmarks, with thorough ablation studies to support the findings.
Weaknesses﻿
1.The AwGA metric currently depends on grounding masks and is only evaluated on the relatively small eGQA dataset (~10k questions). It would be even more reassuring if it could be tested on additional datasets.
2.As noted in the Impact Statement, VLM-based evaluation may inherit biases from the underlying LLM, making it hard to separate errors caused by the OCL representation from those amplified by the LLM.
3.Although mFRESA is presented as just an improved baseline, it performs quite strongly across benchmarks, and its multi-objective reconstruction idea seems interesting enough that a bit more discussion would help readers understand why it works so well.

---

> ### Author Rebuttal · Authors · 2026-03-31
>
> We thank the reviewer for the thoughtful review, positive assessment, and constructive suggestions for improving the paper. Below, we address each concern in turn.
>
> **W1 Small eGQA dataset**: AwGA requires per-question grounding masks, which are unavailable in most VQA datasets (e.g., OK-VQA), making them unsuitable for this evaluation. We therefore use eGQA, which provides high-quality human-verified grounding needed to penalize ``right-answer, wrong-location’’ failures. Although eGQA is relatively small (~10k questions), it is comparable in size to several widely used diagnostic benchmarks, such as SugarCrepe (7.5k), POPE (3k), HallusionBench (1.1k), and MM-Vet (218). Importantly, eGQA is a specialized diagnostic benchmark for grounded reasoning in OCL. We agree that a broader AwGA evaluation would strengthen the paper and will clarify this limitation more explicitly.
>
> **W2 Biases**: Our benchmark measures the downstream usefulness of OCL representations within a shared VLM framework. Since all OCL methods are evaluated with the same LLMs, any LLM bias is largely shared across methods, making the relative comparison informative even if absolute performance is affected. Thus, while absolute scores may inherit some LLM-specific effects (biases), the comparative conclusions across OCL methods remain informative.
>
> **W3 Presentation improvement**: We thank the reviewer for this comment. Due to space considerations, we decided to move mFRESA to the supplemental section. For the camera-ready, we will add a concise main-text description of mFRESA, including its reconstruction targets and why they improve object-centric representations. We discussed the reasons why mFRESA is a better model compared to other OCL methods in Appendix A (L. 612-622). We will move this discussion to the main paper.
>
> **KQ1**: mFRESA incurs a higher one-time training cost, but it provides consistent gains (**≥2 absolute percentage points**) across multiple benchmarks and metrics, including mIoU, accuracy, and AwGA. Since both models were trained on the same hardware (1× A100), we view this as a reasonable performance–compute trade-off. In lower-resource settings, StableLSD may be the more practical choice, whereas mFRESA is preferable when maximizing performance.

---

> > ### Author Rebuttal · Reviewer_WLYs · 2026-04-05
> >
> > Most of my concerns are resolved, thus I will raise my score to 5.

---

> > > ### Author Response · Authors · 2026-04-05
> > >
> > > Thank you for recognizing that the concerns have been fully resolved and for updating your assessment accordingly. We appreciate your constructive feedback, which helped strengthen our manuscript.

---

### Decision · Program_Chairs · 2026-04-30

**Decision:**

Accept (regular)

**Comment:**

This paper argues that common evaluation protocols for object-centric learning (OCL), especially unsupervised object discovery, do not adequately reflect whether slot representations encode useful object properties for diverse downstream tasks. To address this, the authors propose a broader evaluation framework for OCL that (1) leverages vision-instruct-tuned LLMs as VLMs with OCL models as visual encoders for zero-shot evaluation on multiple VQA benchmarks, and (2) introduces Attribution-aware Grounded Accuracy (AwGA), a unified metric combining answer correctness, mask overlap, and gradient-based slot attribution to jointly measure “what” and “where” quality while penalizing fragmentation. In addition, the authors propose mFRESA, a simple multi-target reconstruction baseline (pixels + DINOv2 features + HOG) and report improved results over prior slot-based methods across several tasks.

All viewers reached to a positive consensus on the motivation and framework: “The idea of using instruction-tuned VLMs as zero-shot probes for OCL representations is clever” (Reviewer WLYs); “The VLM-based evaluation protocol is a practical way to scale evaluation across many benchmarks using a unified interface” (Reviewer v2Dz); “The unified "what + where" evaluation framing (AwGA + eGQA) is a useful direction and makes a clear argument against disjoint evaluation” (Reviewer Dv7p). In addition, the experiments are “solid” (Reviewer WLYs) and “strong” (Reviewer Dv7p).

Some concerns regarding small eGQA dataset, biases inherited from LLMs, novel insights, missing related work, additional results and lack of discussions have been adequately addressed in the rebuttal, which have been acknowledged by the reviewers. All three reviewers are inclined to accept this work with consistent recommendations (5, 5, 5).